# Quantifying the Cost of Learning in Queueing Systems

**Daniel Freund**
MIT
Cambridge, MA 02139
dfreund@mit.edu

**Thodoris Lykouris**
MIT
Cambridge, MA 02139
lykouris@mit.edu

**Wentao Weng**
MIT
Cambridge, MA 02139
wweng@mit.edu

## Abstract

Queueing systems are widely applicable stochastic models with use cases in communication networks, healthcare, service systems, etc. Although their optimal control has been extensively studied, most existing approaches assume perfect knowledge of the system parameters. Of course, this assumption rarely holds in practice where there is parameter uncertainty, thus motivating a recent line of work on bandit learning for queueing systems. This nascent stream of research focuses on the asymptotic performance of the proposed algorithms.

In this paper, we argue that an asymptotic metric, which focuses on late-stage performance, is insufficient to capture the intrinsic statistical complexity of learning in queueing systems which typically occurs in the early stage. Instead, we propose the *Cost of Learning in Queueing (CLQ)*, a new metric that quantifies the maximum increase in time-averaged queue length caused by parameter uncertainty. We characterize the CLQ of a single-queue multi-server system, and then extend these results to multi-queue multi-server systems and networks of queues. In establishing our results, we propose a unified analysis framework for CLQ that bridges Lyapunov and bandit analysis, provides guarantees for a wide range of algorithms, and could be of independent interest.[1]

## 1 Introduction

Queueing systems are widely used stochastic models that capture congestion when services are limited. These models have two main components: jobs and servers. Jobs wait in queues and have different types. Servers differ in capabilities and speed. For example, in content moderation of online platforms [28], jobs are user posts with types defined by contents, languages and suspected violation types; servers are human reviewers who decide whether a post is benign or harmful. Moreover, job types can change over time upon receiving service. For instance, in a hospital, patients and doctors can be modeled as jobs and servers. A patient in the queue for emergent care can become a patient in the queue for surgery after seeing a doctor at the emergency department [3]. That is, queues can form a network due to jobs transitioning in types. Queueing systems also find applications in other domains such as call centers [16], communication networks [36] and computer systems [17].

The single-queue multi-server model is a simple example to illustrate the dynamics and decisions in queueing systems. In this model, there is one queue and K servers operating in discrete periods. In each period, a new job arrives with probability $\lambda$. Servers have different service rates $\mu_1, \ldots, \mu_K$. The decision maker (DM) selects a server to serve the first job in the queue if there is any. If server $j$ is selected, the first job in the queue leaves with probability $\mu_j$. The DM aims to minimize the average wait of each job, which is equivalent to minimizing the queue length. The optimal policy thus selects the server with the highest service rate $\mu^\star = \max_j \mu_j$; the usual regime of interest is one where the system is *stabilizable*, i.e., $\mu^\star > \lambda$, which ensures that the queue length does not grow to

---

[1]A full version of this paper [12] can be found at `https://arxiv.org/abs/2308.07817v2`.

37th Conference on Neural Information Processing Systems (NeurIPS 2023).

infinity under an optimal policy. Of course, this policy requires perfect knowledge of the service rates. Under parameter uncertainty, the DM must balance the trade-off between exploring a server with an uncertain rate or exploiting a server with the highest observed service rate.

A recent stream of work studies efficient learning algorithms for queueing systems. First proposed by [41], and later used by [24, 38], *queueing regret* is a common metric to evaluate learning efficiency in queueing systems. In the single-queue multi-server model, let $Q(T, \pi)$ and $Q^\star(T)$ be the number of jobs in period $T$ under a policy $\pi$ and under the optimal policy respectively. Queueing regret is defined as either the last-iterate difference in expected queue length, i.e., $\mathbb{E}\left[Q(T, \pi) - Q^\star(T)\right]$[41, 24], or the time-average version $\frac{1}{T}\mathbb{E}\left[\sum_{t=1}^{T} Q(t, \pi) - Q^\star(t)\right]$ [38]. In the stabilizable case ($\lambda < \mu^\star$), the goal is to bound its scaling relative to the time horizon; examples include $o(T)$ [41], $\tilde{O}(1/T)$ [24], and $O(1/T)$ [38].

In this paper, we argue that an asymptotic metric for per-period queue length does not capture the statistical complexity of learning in queueing systems. This is because, in a queueing system, learning happens in initial periods while queueing regret focuses only on late periods. Whereas cumulative regret in multi-armed bandits is non-decreasing in $T$, the difference in queue lengths between a learning policy and the benchmark optimal policy eventually decreases since the policy eventually learns the parameters (see Figure 1). This leads to the two main questions of this paper:

*1. What metric characterizes the statistical complexity of learning in queueing systems?*

*2. What are efficient learning algorithms for queueing systems?*

Our work studies these questions in general queueing systems that go beyond the single-queue multi-server model and can capture settings such as the hospital and content moderation examples.

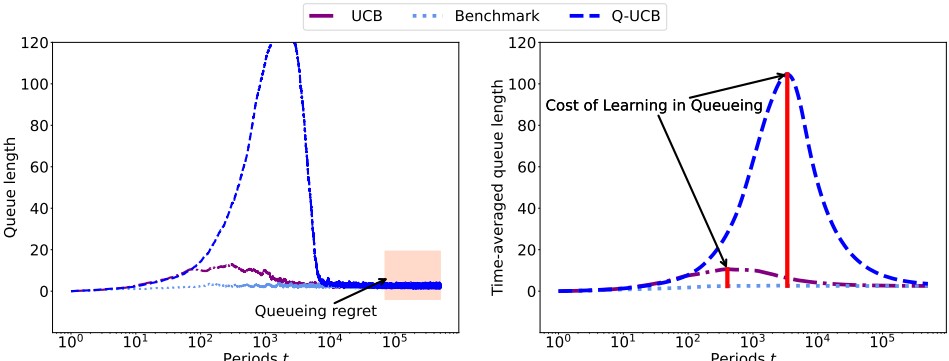

Figure 1: Expected per-period and time-averaged queue lengths of UCB and Q-UCB [24] in a single-queue setting with $K = 5, \lambda = 0.45, \boldsymbol{\mu} = (0.045, 0.35, 0.35, 0.35, 0.55)$; results are averaged over 50 runs. The difference between both algorithms' queue lengths is indistinguishable asymptotically (left figure) though they clearly differ in their learning efficiency for early periods as illustrated by *Cost of Learning in Queueing*, the metric that our work introduces (right figure).

**Cost of learning in queueing.** Tackling the first question, we propose the *Cost of Learning in Queueing* (CLQ) to capture the efficiency of a learning algorithm in queueing systems. The CLQ of a learning policy is defined as the maximum difference of its time-averaged queue length and that of any other policy (with knowledge of parameters) over the entire horizon (see Fig 1 on the right). In contrast to queueing regret, CLQ is 1) a finite-time metric that captures the learning efficiency in early periods and 2) focused on time-averaged queue length instead of per-period queue length. This is favorable as for any periods $1, \ldots, T$, the time-averaged queue length is related to the average wait time by Little's Law. The formal definition of CLQ can be found in Section 3.

**Lower bound of CLQ (Theorem 1).** To characterize the statistical complexity of learning in queueing systems, we consider the simplest non-trivial stabilizable setting that involves one queue and $K$ servers. It is known that the queue length scales as $O(1/\varepsilon)$ under the optimal policy, where $\varepsilon = \mu^\star - \lambda$ is the traffic slackness of the system. Fixing $\varepsilon$ and the number of servers $K$, we establish

a worst-case lower bound $\Omega(\frac{K}{\varepsilon})$ of CLQ. That is, for any $\varepsilon, K$ and a fixed policy, there always exists a setting of arrival and service rates, such that the CLQ of this policy is at least $\Omega(\frac{K}{\varepsilon})$. Combined with the $O(1/\varepsilon)$ optimal queue length, this lower bound result shows that the effect of learning dominates the performance of queueing systems when $K$ is large (as it may increase the maximum time-averaged queue length by a factor of $K$). This is shown in Figure 1 (right) where the peak of time-averaged queue lengths of the optimal policy with knowledge of parameters is much lower than that of the other two policies (our Algorithm 1 and Q-UCB [24]).

**An efficient algorithm for single-queue multi-server systems (Theorem 2).** Given the above lower bound, we show that the Upper Confidence Bound (UCB) algorithm attains an optimal CLQ up to a logarithmic factor in the single-queue multi-server setting. Our analysis is motivated by Fig. 1 (right) where the time-averaged queue length initially increases and then decreases. Based on this pattern, we divide the horizon into an initial *learning* stage and a later *regenerate* stage.

In the learning stage, the queue length increases similar to the multi-armed bandit regret. We formalize this observation by coupling the queue under a policy $\pi$ with a nearly-optimal queue and show that their difference is captured by the *satisficing regret* of $\pi$. Satisficing regret resembles the classical multi-armed bandit regret but disregards the loss of choosing a nearly optimal server (see Eq. (6)); this concept is studied from a Bayesian perspective in multi-armed bandits [32]. Nevertheless, our result in the learning stage is not sufficient as the satisficing regret eventually goes to infinity.

In the regenerate stage, queue lengths decrease as the policy has learned the parameters sufficiently well; the queue then behaves similarly as under the optimal policy and stabilizes the system. To capture this observation, we use Lyapunov analysis and show that the time-averaged queue length for the initial $T$ periods scales as the optimal queue length, but with an additional term depending on the second moment of satisficing regret divided by $T$. Hence, as $T$ increases, the impact of learning gradually disappears. Combining the results in the learning and regenerate stages, we obtain a tight CLQ bound of UCB for the single-queue multi-server setting.

**Efficient algorithms for queueing networks (Theorem 3)** We next generalize the above result to queueing networks that include multiple queues, multiple servers, and transitions of served jobs from servers to queues. For this setting, we build on the celebrated BACKPRESSURE policy that stabilizes a queueing network with knowledge of system parameters [39]. We propose BACKPRESSURE-UCB as a new algorithm to transform BACKPRESSURE into a learning algorithm with appropriate estimates for system parameters and show that its cost of learning scales near-optimally as $\tilde{O}(1/\varepsilon)$ with respect to the traffic slackness $\varepsilon$ (Definition 2). This result extends our framework for single-queue multi-server settings through a coupling approach that reduces the loss incurred by learning in a high-dimensional vector of queue-lengths to a scalar-valued potential function. To the best of our knowledge, this is the first efficient learning algorithm for general queueing networks (see related work for a discussion).

**Related work** A recent line of work studies online learning in queueing systems [40]. To capture uncertainty in services, [41] studies a single-queue setting in which the DM selects a mode of service in each period and the job service time varies between modes (the dependence is a priori unknown and revealed to the DM after the service). The metric of interest is the *queueing regret*, i.e., the difference of queue length between an algorithm and the optimal policy, for which the authors show a sublinear bound. [24] considers the same single-queue multi-server setting as ours and show that a forced exploration algorithm achieves a queueing regret scaling of $\tilde{O}(1/T)$ (under strong structural assumptions this result extends to multiple queues). [38] shows that by probing servers when the queue is idle, it is possible to give an algorithm with queueing regret converging as $O(1/T)$. However, with respect to the traffic slackness $\varepsilon \to 0^+$, both bounds yield suboptimal CLQ: [24] gives at least $O(1/\varepsilon^2)$ and [38] gives at least $O(1/\varepsilon^4)$ (see Appendices A.1, A.2 in [12]). In the analysis of [24], forced exploration is used for low adaptive regret, i.e., regret over any interval [19]; no such guarantee is known for adaptive exploration. But as noted by our Figure 1 and [24, Figure 2], an adaptive exploration algorithm like UCB has a better early-stage performance than Q-UCB. Using our framework in Section 4, we show that UCB indeed has a near-optimal CLQ that scales as $O(\frac{K}{\varepsilon} \ln \frac{K}{\varepsilon})$. Our framework also allows us to show that Q-UCB enjoys a CLQ scaling as $O(\frac{K}{\varepsilon} \ln \frac{K}{\varepsilon} + \ln^3 \frac{K}{\varepsilon})$ (Appendix A.1 in [12]). This improves the guarantee implied by [24] and

shows the inefficiency due to forced exploration is about $O(\ln^3 \frac{K}{\varepsilon})$ and that Q-UCB has both strong transient (CLQ) and asymptotic (queueing regret) performance.

Focusing on the scaling of queueing regret, [23] and [43] study the scheduling in multi-queue settings (with [43] also considering job abandonment), [9, 14] study learning for a load balancing model, [8] studies pricing and capacity sizing for a single-queue single-server model with unknown parameters. For more general settings, [1] designs a Bayesian learning algorithm for Markov Decision Processes with countable state spaces (of which queueing systems are special cases) where parameters are sampled from a known prior over a restricted parameter space; in contrast, our paper does not assume any prior of the unknown parameters. The main difference between all of these works and ours is that we focus on how the maximum time-averaged queue lengths scales with respect to system parameters (traffic slackness and number of queues and servers), not on how the queue lengths scale as time grows. Apart from the stochastic learning setting we focus on, there are works that tackle adversarial learning in queueing systems [21, 26]; these require substantially different algorithms and analyses.

Going beyond queueing regret, there are papers focusing on finite-time queue length guarantees. In a multi-queue multi-server setting, it is known that the MaxWeight algorithm has a polynomial queue length for stabilizable systems. However, it requires knowledge of system parameters. For a joint scheduling and utility maximization problem, [30] combines MaxWeight with forced exploration to handle parameter uncertainty. By selecting a suitable window for sample collection, their guarantee corresponds to a CLQ bound of at least $O(K^4/\varepsilon^3)$ for our single-queue setting (see Appendix A.3 in [12]). [37] studies a multi-queue multi-server setting and propose a frame-based learning algorithm based on MaxWeight. They focus on a greedy approximation which has polynomial queue lengths when the system is stabilizable with twice of the arrival rates. [42] considers a non-stationary setting and shows that combining MaxWeight with discounted UCB estimation leads to stability and polynomial queue length that scales as $\tilde{O}(1/\varepsilon^3)$ (Appendix A.4 in [12]). There is also a line of work studying decentralized learning in multi-queue multi-server settings. [15] assumes queues are selfish and derives conditions under which a no-regret learning algorithm is stable; this is generalized to queueing networks in which queues and servers form a directed acyclic graph by [13]. [33] allows collaborative agents and gives an algorithm with maximum stability, although the queue length scales exponentially in the number of servers. [11] designs a decentralized learning version of MaxWeight and shows that the algorithm always stabilizes the system with polynomial queue lengths $\tilde{O}(1/\varepsilon^3)$ (Appendix A.5 in [12]). In contrast to the above, our work shows for the centralized setting that MaxWeight with UCB achieves the near-optimal time-averaged queue length guarantee of $\tilde{O}(1/\varepsilon)$.

Our paper extends the ability of online learning to general single-class queueing networks [7]. The literature considers different complications that arise in these settings, including jobs of different classes and servers that give service simultaneously to different jobs [39, 10, 7]. For the class of networks we consider, it is known that BACKPRESSURE can stabilize the system with knowledge of system parameters [39]. Noted in [7], one potential drawback of BACKPRESSURE is its need of full knowledge of job transition probabilities. In this regard, our paper contributes to the literature by proposing the first BACKPRESSURE-based algorithm that stabilizes queueing networks without knowledge of system parameters.

Moving beyond our focus on uncertainties in services, an orthogonal line of work studies uncertainties in job types. [2] considers a single server setting where an arriving job belongs to one of two types; but the true type is unknown and is learned by services. They devise a policy that optimizes a linear function of the numbers of correctly identified jobs and the waiting time. [6] studies a similar setting with two types of servers where jobs can route from one server to the others. They focus on the impact on stability due to job type uncertainties. [29, 34] consider multiple job types and server types. Viewing Bayesian updates as job type transitions, they use queueing networks to model the job learning process and give stable algorithms based on BACKPRESSURE. [22, 20] consider online matchings between jobs with unknown payoffs and servers where the goal is to maximize the total payoffs subject to stability. As noted in [29, 34, 22], one key assumption of this line of work is the perfect knowledge of server types and transition probability. Our result thus serves as a step to consider both server uncertainties and job uncertainties, at least in a context without payoffs.

Concurrently to our work, [31] proposes a frame-based MaxWeight algorithm with sliding-window UCB for scheduling in a general multi-queue multi-server system with non-stationary service rates. With a suitable frame size (depending on the traffic slackness), they show stability of the algorithm and obtain a queue length bound of $\tilde{O}(1/\varepsilon^3)$ in the stationary setting (Appendix A.6 in [12]).

## 2 Model

We consider a sequential learning setting where a decision maker (DM) repeatedly schedules jobs to a set of servers of unknown quality over discrete time periods $t = 1, 2, \ldots$. For any $T$, we refer to the initial $T$ periods as the time horizon $T$. To ease exposition, we first describe the simpler setting where there is only one job type (queue) and subsequently extend our approach to a general setting with multiple queues that interact through a network structure.

**Single-queue multi-server system.** A single-queue multi-server system is specified by a tuple $(\mathcal{K}, \lambda, \boldsymbol{\mu})$. There is one queue of jobs and a set of servers $\mathcal{K}$ with $|\mathcal{K}| = K$. The arrival rate of jobs is $\lambda$, that is, in each period there is a probability $\lambda$ that a new job arrives to the queue. The service rate of a server $k \in \mathcal{K}$, that is, the probability it successfully serves the job it is scheduled to work on, is $\mu_k$. Let $Q(t)$ be the number of jobs at the start of period $t$. Initially there is no job and $Q(1) = 0$.

Figure 2 summarizes the events that occur in each period $t$. If there is no job in the queue, i.e., $Q(t) = 0$, then the DM selects no server; to ease notation, they select the null server $J(t) = \perp$.[2] Otherwise, the DM selects a server $J(t) \in \mathcal{K}$ and requests service for the first job in the queue. The

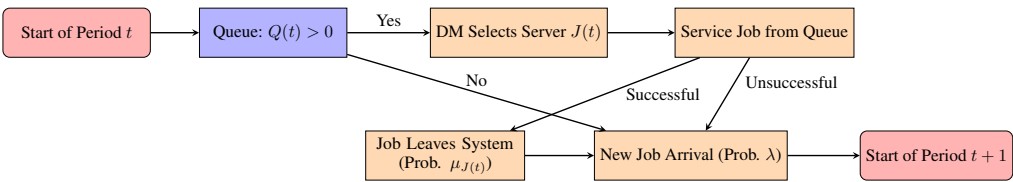

Figure 2: Flowchart for a single-queue multi-server system.

service request is successful with probability $\mu_{J(t)}$ and the job then leaves the system; otherwise, it remains in the queue. At the end of the period, a new job arrives with probability $\lambda$. We assume that arrival and service events are independent. Let $A(t)$ and $\{S_k(t)\}_{k \in \mathcal{K}}$ be a set of independent Bernoulli random variables such that $\mathbb{E}[A(t)] = \lambda$ and $\mathbb{E}[S_k(t)] = \mu_k$ for $k \in \mathcal{K}$; for the null server, $\mu_\perp = S_\perp(t) = 0$ for all $t$. The queue length dynamics are thus given by

$$Q(t+1) = Q(t) - S_{J(t)}(t) + A(t).$$

A non-anticipatory policy $\pi$ for the DM maps for every period $t$ the historical observations until $t$, i.e., $(A(\tau), S_{J(\tau)}(\tau))_{\tau < t}$, to a server $J(t) \in \mathcal{K} \cup \{\perp\}$. We define $Q(t, \pi)$ as the queue length in period $t$ under policy $\pi$. The DM's goal is to select a non-anticipatory policy $\pi$ such that for any time horizon $T \geq 1$, the expected time-averaged queue length $\frac{1}{T} \sum_{t \leq T} \mathbb{E}[Q(t, \pi)]$ is as small as possible.

When service rates are known, the policy $\pi^\star$ selecting the server with the highest service rate $\mu^\star$ in every period (unless the queue is empty) minimizes the expected time-averaged queue length for any time horizon [24, 38]. If $\lambda \geq \mu^*$, even under $\pi^\star$, the expected time-averaged queue length goes to infinity as the time horizon increases. We thus assume $\lambda < \mu^\star$, in which case, the system is *stabilizable*, i.e., the expected time-averaged queue length under $\pi^\star$ is bounded by a constant over the entire time horizon. We next define the *traffic slackness* of this system:

**Definition 1.** *A single-queue multi-server system has a traffic slackness* $\varepsilon \in (0, 1]$ *if* $\lambda + \varepsilon \leq \mu^\star$.

A larger traffic slackness implies that a system is easier to stabilize. It is known that the policy $\pi^\star$ obtains an expected time-averaged queue length of the order of $\frac{1}{\varepsilon}$ [36].

**Queueing network.** A queueing network extends the above case by having multiple queues and probabilistic job transitions after service completion; our model here resembles the one in [7]. A queueing network is defined by a tuple $(\mathcal{N}, \mathcal{K}, \boldsymbol{\Lambda}, \boldsymbol{\mu}, \boldsymbol{\mathcal{A}}, \boldsymbol{\Sigma}, \boldsymbol{\mathcal{B}}, \boldsymbol{\mathcal{D}}, \boldsymbol{P})$, where $\boldsymbol{\mathcal{B}} = \{\mathcal{B}_n\}_{n \in \mathcal{N}}$, $\boldsymbol{\mathcal{D}} = \{\mathcal{D}_k\}_{k \in \mathcal{K}}$ and $\boldsymbol{P} = (p_{k,n})_{k \in \mathcal{K}, n \in \mathcal{N} \cup \{\perp\}}$ such that $\sum_{n \in \mathcal{N} \cup \{\perp\}} p_{k,n} = 1, \forall k$. In contrast to the single-queue case, there is now a set of queues $\mathcal{N}$ with cardinality $N$ and a virtual queue $\perp$ to which jobs transition once they leave the system. Each queue $n \in \mathcal{N}$ has a set of servers $\mathcal{B}_n$, each of which

---

[2]Compared to common assumptions in the literature, e.g., [38, 11, 42], this makes for a more challenging setting as algorithms cannot learn service rates by querying servers in periods when they have no jobs.

belongs to a single queue. As before, the service rate of server $k$ is $\mu_k$. The set $\mathcal{D}_k$ contains the destination queues of server $k$ (and can include the virtual queue $\perp$).

In each period $t$, the DM selects a set of servers to schedule jobs to and, if a service request from queue $n$ to server $k$ is successful, a job from queue $n$ transitions to a queue $n' \in \mathcal{D}_k$ with probability $p_{k,n'}$ (this implies $p_{k,n'} = 0$ if $n' \notin \mathcal{D}_k$). The selected set of servers comes from a set of feasible schedules $\boldsymbol{\Sigma} \subseteq \{0,1\}^{\mathcal{K}}$, which captures interference between servers. We require that for any queue, the number of selected servers is no larger than the number of jobs in this queue.[3] Formally, letting $\sigma_k = 1$ if schedule $\boldsymbol{\sigma} \in \boldsymbol{\Sigma}$ selects server $k$ and denoting $\boldsymbol{Q}(t) = (Q_n(t))_{n \in \mathcal{N}}$ as the queue length vector at the beginning of period $t$, the set of feasible schedules in this period is

$$\boldsymbol{\Sigma}_t = \{\boldsymbol{\sigma} \in \boldsymbol{\Sigma} \colon \sum_{k \in \mathcal{B}_n} \sigma_k \leq Q_n(t), \forall n \in \mathcal{N}\}$$

and the DM's decision in period $t$ is to select a schedule $\boldsymbol{\sigma}(t) \in \boldsymbol{\Sigma}_t$. Following [39], we assume that any subset of a feasible schedule is still feasible, i.e., if $\boldsymbol{\sigma} \in \boldsymbol{\Sigma}$ and $\sigma'_k \leq \sigma_k \; \forall k \in \mathcal{K}$, then $\boldsymbol{\sigma}' \in \boldsymbol{\Sigma}$.

We now formalize the arrival and service dynamics in every period, which are captured by the independent random variables $\{\boldsymbol{A}(t), \{S_k(t)\}_{k \in \mathcal{K}}\}_t$. The arrival vector $\boldsymbol{A}(t) = \{A_n(t)\}_{n \in \mathcal{N}}$ consists of (possibly correlated) random variables $A_n(t)$ taking value in $\mathcal{A} \subseteq \{0,1\}^{\mathcal{N}}$; we denote its distribution by $\boldsymbol{\Lambda}$ and let $\mathbb{E}[A_n(t)] = \lambda_n(\boldsymbol{\Lambda})$ with $\boldsymbol{\lambda} = (\lambda_n(\boldsymbol{\Lambda}))_{n \in \mathcal{N}}$. The service $S_k(t)$ for each server $k \in \mathcal{K}$ is a Bernoulli random variable indicating whether the selected service request was successful.[4] To formalize the job transition, let $\boldsymbol{L}_k(t) = (L_{k,n}(t))_{n \in \mathcal{N} \cup \{\perp\}}$ be a random vector over $\{0,1\}^{\mathcal{N} \cup \{\perp\}}$ for server $k$ independent of other randomness such that $\mathbb{P}\{L_{k,n}(t) = 1\} = p_{k,n}$ and $\sum_{n \in \mathcal{N} \cup \{\perp\}} L_{k,n}(t) = 1$. The queueing dynamic is given by

$$Q_n(t+1) = Q_n(t) - \sum_{k \in \mathcal{B}_n} \sigma_k(t) S_k(t) + A_n(t) + \sum_{k' \in \mathcal{K}} \sigma_{k'}(t) S_{k'}(t) L_{k',n}(t). \tag{1}$$

We assume that the DM has knowledge of which policies are allowed, i.e., they know $\boldsymbol{\Sigma}$ and $\mathcal{B}$, but has no prior knowledge of the rates $\boldsymbol{\lambda}, \boldsymbol{\mu}, \boldsymbol{P}$ and the set $\mathcal{D}$. In period $t$, the observed history is the set $\left(\{A_n(\tau)\}_{n \in \mathcal{N}}, \{S_k(\tau) L_{k,n}(\tau)\}_{n \in \bar{\mathcal{N}}, k \in \mathcal{K} \colon \sigma_k(\tau) = 1}\right)_{\tau < t}$ that includes transition information on top of arrivals and services. Note that a job transition is only observed when the server is selected and the service is successful. Similar to before, a non-anticipatory policy $\pi$ maps an observed history to a feasible schedule; we let $Q_n(t, \pi)$ be the length of queue $n$ in period $t$ under this policy.

Unlike the single-queue case, it is usually difficult to find the optimal policy for a queueing network even with known system parameters. Fortunately, if the system is stabilizable, i.e., $\lim_{t \to \infty} \frac{1}{T} \sum_{t \leq T} \mathbb{E}[\|\boldsymbol{Q}(t)\|_1] < \infty$ under some scheduling policy, then the arrival rate vector must be within the *capacity region* of the servers [39]. Formally, let $\boldsymbol{\Phi} = \{\boldsymbol{\phi} \in [0,1]^{\boldsymbol{\Sigma}} \colon \sum_{\boldsymbol{\sigma} \in \boldsymbol{\Sigma}} \phi_{\boldsymbol{\sigma}} = 1\}$ be the probability simplex over $\boldsymbol{\Sigma}$. A distribution $\boldsymbol{\phi}$ in $\boldsymbol{\Phi}$ can be viewed as the frequency of a policy using each schedule $\boldsymbol{\sigma} \in \boldsymbol{\Sigma}$, and the effective service rate queue $n$ can get is given by $\mu_n^{\text{net}}(\boldsymbol{\phi}) = \sum_{\boldsymbol{\sigma} \in \boldsymbol{\Sigma}} \phi_{\boldsymbol{\sigma}} \left(\sum_{k \in \mathcal{B}_n} \sigma_k \mu_k - \sum_{k' \in \mathcal{K}} \sigma_{k'} \mu_{k'} p_{k',n}\right)$; this includes both job inflow and outflow. We denote the effective service rate vector for a schedule distribution $\boldsymbol{\phi}$ by $\boldsymbol{\mu}^{\text{net}}(\boldsymbol{\phi})$. Then, the capacity region is $\mathcal{S}(\boldsymbol{\mu}, \boldsymbol{\Sigma}, \mathcal{B}, \boldsymbol{P}) = \{\boldsymbol{\mu}^{\text{net}}(\boldsymbol{\phi}) \colon \boldsymbol{\phi} \in \boldsymbol{\Phi}\}$. For a queueing network to be stabilizable, we must have $\boldsymbol{\lambda}(\boldsymbol{\Lambda}) \in \mathcal{S}(\boldsymbol{\mu}, \boldsymbol{\Sigma}, \mathcal{B}, \boldsymbol{P})$ [39]. As in the single-queue case, we further assume that the system has a positive traffic slackness and let $\mathbf{1}$ denote a vector of 1s with suitable dimension.

**Definition 2.** *A queueing network has traffic slackness* $\varepsilon \in (0, 1]$ *if* $\boldsymbol{\lambda}(\boldsymbol{\Lambda}) + \varepsilon \mathbf{1} \in \mathcal{S}(\boldsymbol{\mu}, \boldsymbol{\Sigma}, \mathcal{B}, \boldsymbol{P})$.

We also study a special case of queueing networks, multi-queue multi-server systems, where jobs immediately leave after a successful service, i.e., $\mathcal{D}_k = \{\perp\}$ for all $k \in \mathcal{K}$; this extends the models in [11, 42] mentioned above. Since the transition probability matrix $\boldsymbol{P}$ is trivial ($p_{k,\perp} = 1, \forall k$), we denote the capacity region of a multi-queue multi-server system by $\mathcal{S}(\boldsymbol{\mu}, \boldsymbol{\Sigma}, \mathcal{B})$.

---

[3]Though this reflects the feature from the single-queue setting, that $Q(t) = 0 \implies J(t) = \perp$, it maintains the flexibility to have a queue that has multiple jobs served in a single period.

[4]This formulation captures settings where arrivals are independent of the history, such as the example of each queue having independent arrivals (e.g., the bipartite queueing model in [11]) and the example of feature-based queues [35] where jobs have features; each type of feature has one queue; at most one job arrives among all queues in each period. Our formulation cannot capture state-dependent arrivals such as queues with balking [18].

# 3 Main results: the statistical complexity of learning in queueing

This section presents our main results on the statistical complexity of learning in queueing systems. We first define the *Cost of Learning in Queueing*, or CLQ as a shorthand, a metric capturing this complexity and provide a lower bound for the single-queue multi-server setting. Motivated by this, we design an efficient algorithm for the single-queue multi-server setting with a matching CLQ and then extend this to the multi-queue multi-server and queueing network systems.

## 3.1 Cost of Learning in Queueing

We first consider learning in the single-queue multi-server setting. Previous works on learning in queueing systems focus on the queueing regret $\mathbb{E}\left[Q(T, \pi) - Q(T, \pi^\star)\right]$ in the asymptotic regime of $T \to \infty$. The starting point of our work stems from the observation that an asymptotic metric, which measures performance in late periods, cannot capture the complexity of learning as learning happens in initial periods (recall the left of Figure 1). In addition, a guarantee on per-period queue length cannot easily translate to the service experience (or wait time) of jobs.

Motivated by the above insufficiency of queueing regret, we define the *Cost of Learning in Queueing* (or CLQ) as the maximum increase in expected time-averaged queue lengths under policy $\pi$ compared with the optimal policy. Specifically, we define the single-queue CLQ as:

$$\text{CLQ}^{\text{single}}(\lambda, \boldsymbol{\mu}, \pi) = \max_{T \geq 1} \frac{\sum_{t=1}^{T} \mathbb{E}\left[Q(t, \pi) - Q(t, \pi^\star)\right]}{T}. \tag{2}$$

As shown in Figure 1 (right), CLQ is a finite-time metric and explicitly takes into account how fast learning occurs in the initial periods. In addition, a bound on the maximum increase in time-averaged queue length translates approximately (via Little's Law [27]) to a bound on the increase in average job wait times.

Given that the traffic slackness measures the difficulty of stabilizing a system, we also consider the worst-case cost of learning in queueing over all pairs of $(\lambda, \boldsymbol{\mu})$ with a fixed traffic slackness $\varepsilon$. In a slight abuse of notation, we overload $\text{CLQ}^{\text{single}}$ to also denote this worst-case value, i.e.,

$$\text{CLQ}^{\text{single}}(K, \varepsilon, \pi) = \sup_{\lambda \in [0,1), \boldsymbol{\mu} \in [0,1]^K : \, \lambda + \varepsilon \leq \max_k \mu_k} \text{CLQ}^{\text{single}}(\lambda, \boldsymbol{\mu}, \pi). \tag{3}$$

Our goal is to design a policy $\pi$, without knowledge of the arrival rate, the service rates, and the traffic slackness, that achieves low worst-case cost of learning in a single-queue multi-server system.

We can extend the definition of CLQ to the multi-queue multi-server and the queueing network settings. Since the optimal policy is difficult to design, we instead define CLQ for a policy $\pi$ by comparing it with any non-anticipatory policy (which makes decisions only based on the history):

$$\text{CLQ}^{\text{multi}}(\boldsymbol{\Lambda}, \boldsymbol{\mu}, \boldsymbol{\Sigma}, \boldsymbol{\mathcal{B}}, \pi) = \max_{\text{non-anticipatory } \pi'} \max_{T \geq 1} \frac{\sum_{t=1}^{T} \sum_{n \in \mathcal{N}} \mathbb{E}\left[Q_n(t, \pi) - Q_n(t, \pi')\right]}{T}. \tag{4}$$

$$\text{CLQ}^{\text{net}}(\boldsymbol{\Lambda}, \boldsymbol{\mu}, \boldsymbol{\Sigma}, \boldsymbol{\mathcal{B}}, \boldsymbol{\mathcal{D}}, \boldsymbol{P}, \pi) = \max_{\text{non-anticipatory } \pi'} \max_{T \geq 1} \frac{\sum_{t=1}^{T} \sum_{n \in \mathcal{N}} \mathbb{E}\left[Q_n(t, \pi) - Q_n(t, \pi')\right]}{T} \tag{5}$$

As in the single-queue setting, we can define the worst-case cost of learning for a fixed structure $\mathcal{A}, \boldsymbol{\Sigma}, \boldsymbol{\mathcal{B}}, \boldsymbol{\mathcal{D}}$ and a traffic slackness $\varepsilon$ as the supremum across all arrival, service, and transition rates with this traffic slackness. With the same slight abuse of notation as before, we denote these quantities by $\text{CLQ}^{\text{multi}}(\mathcal{A}, \boldsymbol{\Sigma}, \boldsymbol{\mathcal{B}}, \varepsilon, \pi)$ and $\text{CLQ}^{\text{net}}(\mathcal{A}, \boldsymbol{\Sigma}, \boldsymbol{\mathcal{B}}, \boldsymbol{\mathcal{D}}, \varepsilon, \pi)$.

## 3.2 Lower bound on the cost of learning in queueing

Our first result establishes a lower bound on $\text{CLQ}^{\text{single}}(K, \varepsilon, \pi)$. In particular, for any feasible policy $\pi$, we show a lower bound of $\Omega(\frac{K}{\varepsilon})$ for sufficiently large $K$. With known parameters, the optimal time-averaged queue length is of the order of $1/\varepsilon$. Hence, our result shows that the cost of learning is non-negligible in queueing systems when there are many servers. For fixed $K$ and $\varepsilon$, our lower bound considers the cost of learning of the worst-case setting and is instance-independent.

**Theorem 1.** *For $K \geq 2^{14}, \varepsilon \in (0, 0.25]$ and feasible policy $\pi$, we have* $\text{CLQ}^{\text{single}}(K, \varepsilon, \pi) \geq \frac{K}{2^{14}\varepsilon}$.

Although our proof is based on the distribution-free lower bound $\Omega(\sqrt{KT})$ for classical multi-armed bandits [5], this result does not apply directly to our setting. In particular, suppose the queue in our system is never empty. Then the accumulated loss in service of a policy is exactly the *regret* in bandits and the lower bound implies that any feasible policy serves at least $\Omega(\sqrt{KT})$ jobs fewer than the optimal policy in the first $T$ periods. However, due to the traffic slackness, the queue does get empty under the optimal policy, and in periods when this occurs, the optimal policy also does not receive service. As a result, the queue length of a learning policy could be lower than $\Theta(\sqrt{KT})$ despite the loss of service compared with the optimal policy.

We next discuss the intuition (formal proof in Appendix C.1 of [12]). Fixing $K$ and $\varepsilon$, suppose the gap in service rates between the optimal server and others is $2\varepsilon$. Then for any $t$ in a time horizon $T = O(\frac{K}{\varepsilon^2})$, the number of arrivals in the first $t$ periods is around $\lambda t$ and the potential service of the optimal server is around $(\lambda + \varepsilon)t$. By the multi-armed bandit lower bound, the total service of a policy is at most around $\lambda t + \varepsilon t - \sqrt{Kt} \leq \lambda t$ since $t \leq T = O(\frac{K}{\varepsilon^2})$. Then the combined service rate of servers chosen in the first $T$ periods, i.e., $\sum_{t \leq T} \mu_{J(t)}$, is strictly bounded from above by the total arrival rate $\lambda t$. A carefully constructed example shows that the number of unserved jobs is around $\varepsilon t$ for every $t \leq T$ and thus the time-averaged queue length for the horizon $T$ is of the order of $\varepsilon T = O(\frac{K}{\varepsilon})$.

**Remark 1.** *In [24, Proposition 3], the authors established an instance-dependent lower bound on $\mathbb{E}[Q(t,\pi) - Q^\star(t)]$. The implied CLQ lower bound is weaker than ours ($\tilde{\Omega}(K)$, see Appendix C.2 in [12]) and is constrained to $\alpha-$consistent policy $\pi$ (see definition 1 in [24]) whereas ours does not.*

### 3.3 Upper bound on the cost of learning in queueing

Motivated by the lower bound, we propose efficient algorithms with a focus on heavy-traffic optimality, i.e., ensuring $\text{CLQ} = \tilde{O}(1/\varepsilon)$ as $\varepsilon \to 0^+$. The rationale is that stabilizing the system with unknown parameters is more difficult when the traffic slackness is lower as an efficient algorithm must strive to learn parameters more accurately. We establish below that the classical upper confidence bound policy (UCB, see Algorithm 1) achieves near-optimal $\text{CLQ} = \tilde{O}(\frac{K}{\varepsilon})$ for any $K$ and $\varepsilon$ in the single-queue multi-server setting with no prior information of any system parameters.

**Theorem 2.** *For any $K \geq 1, \varepsilon \in (0, 1]$, $\text{CLQ}^{\text{single}}(K, \varepsilon, \text{UCB}) \leq \frac{323K + 64K(\ln K + 2\ln 1/\varepsilon)}{\varepsilon}$.*

The proof of the theorem (Section 4) bridges Lyapunov and bandit analysis, makes an interesting connection to *satisficing regret*, and is a main technical contribution of our work.

We next extend our approach to the queueing network setting. Fixing the system structure $\mathcal{A}, \Sigma, \mathcal{B}, \mathcal{D}$, we define $M_{\mathcal{A}} = \max_{\mathbf{A} \in \mathcal{A}} \sum_{n \in \mathcal{N}} A_n$ and $M_{\Sigma, \mathcal{B}} = \max_{\sigma \in \Sigma} \sum_{n \in \mathcal{N}} \sum_{k \in \mathcal{B}_n} \sigma_k$, to be the maximum number of new job arrivals and the maximum number of selected servers per period. Further, for queueing networks, we also define the quantity $M_{\mathcal{D}} = \sum_{k \in \mathcal{K}} |\mathcal{D}_k|^2$, related to the number of queues each server may see its jobs transition to. The following result shows that the worst-case cost of learning of our algorithm BACKPRESSURE-UCB (BP-UCB as a short-hand, [12, Algorithm 3]) has optimal dependence on $\frac{1}{\varepsilon}$. The proof of this Theorem is provided in [12, Section 6].

**Theorem 3.** *For any $\mathcal{A}, \Sigma, \mathcal{B}, \mathcal{D}$ and traffic slackness $\varepsilon \in (0, 1]$, we have*

$$\text{CLQ}^{\text{net}}(\mathcal{A}, \Sigma, \mathcal{B}, \mathcal{D}, \varepsilon, \text{BP-UCB}) \leq \frac{\sqrt{N}\left(32M_{\mathcal{A}} + 2^{12}M_{\mathcal{D}}M_{\Sigma, \mathcal{B}}^2\left(1 + \ln(M_{\mathcal{A}}M_{\mathcal{D}}M_{\Sigma, \mathcal{B}}/\varepsilon)\right)\right)}{\varepsilon}.$$

Our proof builds on the special case of multi-queue multi-server systems, for which we provide Algorithm MAXWEIGHT-UCB with a corresponding performance guarantee (see [12, Section 5]).

## 4 Proof for the single-queue multi-server setting (Theorem 2)

In this section, we bound the CLQ of UCB for the single-queue multi-server setting (Theorem 2). In each period $t$, when the queue is non-empty, UCB selects a server with the highest upper confidence bound estimation $\bar{\mu}_k(t) = \min\left(1, \hat{\mu}_k(t) + \sqrt{\frac{2\ln(t)}{C_k(t)}}\right)$ where $\hat{\mu}_k(t)$ is the sample mean of services and $C_k(t)$ is the number of times server $k$ is selected in the first $t-1$ periods.

---

**Algorithm 1:** UCB for a single-queue multi-server system

---

Sample mean $\hat{\mu}_k(1) \leftarrow 0$, number of samples $C_k(1) \leftarrow 0$ for $k \in \mathcal{K} \cup \{\perp\}$, queue $Q(1) \leftarrow 0$
**for** $t = 1 \ldots$ **do**

$\quad \bar{\mu}_k(t) = \min\left(1, \hat{\mu}_k(t) + \sqrt{\frac{2\ln(t)}{C_k(t)}}\right), \forall k \in \mathcal{K}$

$\quad$ **if** $Q(t) > 0$ **then** $J(t) \leftarrow \arg\max_k \bar{\mu}_k(t);$ **else** $J(t) \leftarrow \perp$

$\quad$ /* Update queue length & estimates based on $S_{J(t)}(t)$, $A(t)$, and $J(t)$     */

1 $\quad Q(t+1) \leftarrow Q(t) - S_{J(t)}(t) + A(t)$

2 $\quad C_{J(t)}(t+1) \leftarrow C_{J(t)}(t) + 1, \quad \hat{\mu}_{J(t)}(t+1) \leftarrow \frac{C_{J(t)}(t)\hat{\mu}_{J(t)}(t) + S_{J(t)}(t)}{C_{J(t)}(t+1)}$

3 $\quad$ **for** $k \neq J(t)$ **set** $C_k(t+1) \leftarrow C_k(t), \quad \hat{\mu}_k(t+1) \leftarrow \hat{\mu}_k(t)$

---

We establish a framework to upper bound $\mathrm{CLQ}^{\mathrm{single}}$ for any policy by considering separately the initial *learning* stage and the later *regenerate* stage. The two stages are separated by a parameter $T_1$ that appears in our analysis: intuitively, during the learning stage ($t < T_1$), the loss in total service of a policy compared with the optimal server's outweighs the slackness $\varepsilon$ of the system (Definition 1), i.e., $\sum_{\tau=1}^{t} \mu^\star - \mu_{J(\tau)} > t\varepsilon$ and thus the queue length grows linearly with respect to the left-hand side. After the learning stage ($t > T_1$), when $\sum_{\tau=1}^{t} \mu^\star - \mu_{J(\tau)} < t\varepsilon$, the queue regenerates to a constant length independent of $t$. To prove the $\tilde{O}(\frac{K}{\varepsilon})$ bound on $\mathrm{CLQ}^{\mathrm{single}}$, we couple the queue with an "auxiliary" queue where the DM always chooses a nearly optimal server in the learning stage. Then we utilize a Lyapunov analysis to bound the queue length during the regenerate stage.

The framework establishes a connection between $\mathrm{CLQ}^{\mathrm{single}}(\lambda, \boldsymbol{\mu}, \pi)$ and the *satisficing regret* defined as follows. For any horizon $T$, the satisficing regret $\mathrm{SaR}^{\mathrm{single}}(\pi, T)$ is the total service rate gap between the optimal server and the server selected by $\pi$ except for the periods where the gap is less than $\frac{\varepsilon}{2}$ or the queue length is zero. That is, the selected server is satisficing as long as its service rate is nearly optimal or the queue is empty. To formally define it, we denote $\max(x, 0)$ by $x^+$ and define the satisficing regret of a policy $\pi$ over the first $T$ periods by

$$\mathrm{SaR}^{\mathrm{single}}(\pi, T) = \sum_{t=1}^{T} \left(\mu^\star - \mu_{J(t)} - \frac{\varepsilon}{2}\right)^+ \mathbb{1}\left(Q(t) \geq 1\right) \tag{6}$$

We use the satisficing regret notation because our motivation is similar to that in multi-armed bandits [32], initially considered for an infinite horizon and a Bayesian setting. In multi-armed bandits, optimal bounds on regret $\sum_{t=1}^{T}(\mu^\star - \mu_{J(t)})$ are either instance-dependent $\mathcal{O}\left(\sum_{k: \mu_k < \mu^\star} \left(\frac{\ln T}{\mu^\star - \mu_k}\right)\right)$ [4] or instance-independent $O(\sqrt{KT})$ [5]. However, both are futile to establish a $\tilde{\mathcal{O}}(\frac{K}{\varepsilon})$ bound for $\mathrm{CLQ}^{\mathrm{single}}(K, \varepsilon)$: The first bound depends on the minimum gap (which can be infinitesimal), whereas the second is insufficient as we explain in the discussion after Lemma 4.2.

We circumvent these obstacles by connecting the time-averaged queue length of the system with the satisficing regret of the policy via Lemma 4.1 (for the learning stage) and Lemma 4.2 (for the regenerate stage). Lemma 4.1 explicitly bounds the expected queue length through the expected satisficing regret; this is useful during the learning stage but does not give a strong bound for the regenerate stage. Lemma 4.2 gives a bound that depends on $\frac{\mathrm{SaR}^{\mathrm{single}}(\pi, T)^2}{T}$, and is particularly useful during the latter regenerate stage. We then show that the satisficing regret of UCB is $O(\frac{K \ln T}{\varepsilon})$ (Lemma 4.3). Combining these results, we establish a tight bound for the cost of learning of UCB.

Formally, Lemma 4.1 shows that the expected queue length under $\pi$ in period $t$ is at most that under a nearly optimal policy plus the expected satisficing regret up to that time.

**Lemma 4.1.** *For any policy $\pi$ and horizon $T$, we have $\frac{\sum_{t=1}^{T} \mathbb{E}[Q(t)]}{T} \leq \frac{3}{\varepsilon} + \mathbb{E}\left[\mathrm{SaR}^{\mathrm{single}}(\pi, T)\right]$.*

Lemma 4.1 is established by coupling the queue with an auxiliary queue that always selects a nearly optimal server. However, it cannot provide a useful bound on the cost of learning. For large $T$, it is known that $\mathbb{E}\left[\mathrm{SaR}^{\mathrm{single}}(\pi, T)\right]$ must grow with a rate of at least $\mathcal{O}(\frac{\log T}{\varepsilon})$ [25]. Hence, Lemma 4.1 only meaningfully bounds the queue length for small $T$ (learning stage). For large $T$ (regenerate stage), we instead have the following bound (Lemma 4.2).

**Lemma 4.2.** *For any policy $\pi$ and horizon $T$, we have* $\frac{\sum_{t=1}^{T} \mathbb{E}[Q(t)]}{T} \leq \frac{4}{\varepsilon} + \frac{8}{\varepsilon^2} \cdot \frac{\mathbb{E}\left[\text{SaR}^{\text{single}}(\pi,T)^2\right]}{T}.$

This lemma shows that the impact of learning, reflected by $\mathbb{E}\left[\text{SaR}^{\text{single}}(\pi,T)^2\right]$, decays at a rate of $\frac{1}{T}$. Therefore, as long as $\text{SaR}^{\text{single}}(\pi,T)^2$ is of a smaller order than $T$, the impact of learning eventually disappears. This also explains why the instance-independent $O(\sqrt{KT})$ regret bound for multi-armed bandits is insufficient for our analysis: the second moment of the regret scales linearly with the horizon and does not allow us to show a decreasing impact of learning on queue lengths.

Lemma 4.2 suffices to show stability ($\lim_{T\to\infty} \frac{\sum_{t=1}^{T} \mathbb{E}[Q(t)]}{T} < \infty$), but gives a suboptimal bound for small $T$. Specifically, when $\text{SaR}^{\text{single}}(\pi,T)^2 \gtrsim T$, this bound is of the order of $\Omega(\frac{1}{\varepsilon^2})$.[5] We thus need both Lemma 4.1 and Lemma 4.2 to establish a tight bound on the cost of learning in queues.

The following result bounds the first and second moments of the satisficing regret of UCB.

**Lemma 4.3.** *For any horizon $T$, we have*

$$(i)\ \mathbb{E}\left[\text{SaR}^{\text{single}}(\text{UCB},T)\right] \leq \frac{16K(\ln T + 2)}{\varepsilon}, \ (ii)\ \mathbb{E}\left[\text{SaR}^{\text{single}}(\text{UCB},T)^2\right] \leq \frac{2^9 K^2 (\ln T + 2)^2}{\varepsilon^2}.$$

We next offer a proof sketch of Theorem 2. The theorem and lemmas are proven in [12, Section 4].

*Proof sketch of Theorem 2.* Fix any pair of $\lambda, \boldsymbol{\mu}$ with $\max_{k\in\mathcal{K}} \mu_k = \lambda + \varepsilon$. Let $T_1 = \left\lfloor \left(2^{12} K^2 / \varepsilon^4\right)^2 \right\rfloor$. We bound $\text{CLQ}^{\text{single}}(\lambda, \boldsymbol{\mu}, \text{UCB})$ by considering $T \leq T_1$ (learning stage) and $T \geq T_1$ (regenerate stage) separately. For $T \leq T_1$, we have

$$\frac{1}{T}\sum_{t\leq T} \mathbb{E}[Q(t)] \leq \frac{3}{\varepsilon} + \mathbb{E}\left[\text{SaR}^{\text{single}}(\text{UCB},T)\right] \leq \frac{323K + 64K(\ln K + 2\ln 1/\varepsilon)}{\varepsilon},$$

where we use Lemmas 4.1 and 4.3 (i). For $T > T_1$, we have

$$\frac{1}{T}\sum_{t\leq T} \mathbb{E}[Q(t)] \leq \frac{4}{\varepsilon} + \frac{8}{\varepsilon^2}\frac{\mathbb{E}\left[\text{SaR}^{\text{single}}(\text{UCB},T)^2\right]}{T} \leq \frac{323K + 64K(\ln K + 2\ln 1/\varepsilon)}{\varepsilon}, \quad (7)$$

where we use Lemmas 4.2 and 4.3 (ii). $\qquad\square$

**Remark 2.** *Although the* CLQ *metric is focused on the entire horizon, our analysis extends to bounding the maximum expected time-averaged queue lengths in the later horizon, which is formalized as* $\max_{T\geq T_1} \frac{\sum_{t\leq T} \mathbb{E}[Q(t)]}{T}$ *for any $T_1$. In particular, for $T_1 \geq \left(\frac{2^{12}K^2}{\varepsilon^4}\right)^2$, Lemma 4.2 shows that* $\max_{T\geq T_1} \frac{\sum_{t\leq T} \mathbb{E}[Q(t)]}{T} \leq \frac{5}{\varepsilon}$; *UCB thus enjoys the optimal asymptotic queue length scaling of $O(\frac{1}{\varepsilon})$.*

## 5   Conclusions

Motivated by the observation that queueing regret does not capture the complexity of learning which tends to occur in the initial stages, we propose an alternative metric (CLQ) to encapsulate the statistical complexity of learning in queueing systems. For a single-queue multi-server system with $K$ servers and a traffic slackness $\varepsilon$, we derive a lower bound $\Omega(\frac{K}{\varepsilon})$ on CLQ, thus establishing that learning incurs a non-negligible increase in queue lengths. We then show that the classical UCB algorithm has a matching upper bound of $\tilde{O}(\frac{K}{\varepsilon})$. Finally, we extend our result to multi-queue multi-sever systems and general queueing networks by providing algorithms, MAXWEIGHT-UCB and BACKPRESSURE-UCB, whose CLQ has a near optimal $\tilde{O}(1/\varepsilon)$ dependence on traffic slackness.

Having introduced a metric that captures the complexity of learning in queueing systems, our work can serve as a starting point for interesting extensions that can help shed further light on the area. In particular, future research may focus on beyond worst case guarantees for CLQ, non-stationary settings, improved bounds using contextual information, etc.

---

[5]This is suboptimal as long as in the second term of Lemma 4.2 we have an exponent greater than 1 for $1/\varepsilon$.

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
