# OpenReview forum: "Quantifying the Cost of Learning in Queueing Systems"
_NeurIPS.cc/2023/Conference — NeurIPS 2023 poster_

### Official Review · Reviewer_ZtnW · 2023-07-01

**Soundness:** 2 fair
**Presentation:** 2 fair
**Contribution:** 2 fair
**Rating:** 4
**Confidence:** 4

**Summary:**

In this paper, the authors introduce a new regret metric, called Cost of Learning in Queueing (CLQ), to quantify the rate at which an optimal scheduling policy can be learned to minimize the time average queue lengths. The authors derive a lower bound to CLQ and show that an UCB-based policy comes close to achieving the lower bound. They also extend their policy to multiple queues in a network setting by combining the UCB policy with the celebrated Backpressure policy.

**Strengths:**

1. The dependence of the regret bound with respect to the slack parameter (epsilon) is optimal.
2. Bounding the CLQ metric in terms of satisficing regret is intriguing.


**Weaknesses:**

1.	The paper [34] considers an unnormalized version of the same metric proposed in this paper. However, the results presented in this paper give a potentially weaker bound (linearly increasing) than the result in [34]. To be more specific, Theorem 2 proves a constant (let’s denote it by $c$) upper bound to the CLQ metric. Plugging this upper bound into Eq (3) yields the following linear bound on the queue length regret:

   $\sum_{t=1}^T \mathbb{E}\left[(Q(t, \pi)) -  Q(t, \pi^*) \right]\leq cT, ~ \forall T \geq 1 $

But it is already known from [34, Theorem 2, 3] that there exist simple dynamic policies under which the queue-length regret (i.e., the LHS of the above equation) can be bounded by a constant. Clearly, the CLQ metric fails to capture this strong result and paints an overly pessimistic picture. It is also not clear if the result presented in this paper strictly and quantitatively improves upon [34], even for smaller horizon lengths.

2.	The proposed algorithm directly uses UCB, and its analysis does not present any new technical insights.


**Questions:**

1.	Please respond to Comment 1 above by arguing why Theorem 2 gives a stronger bound than [34].
2.	Since the proposed UCB-based policy directly estimates the mean service rate, it might not work in the non-stationary setting, although the classic Max-weight (or Backpressure) might be able to stabilize the queues. Can the authors shed light on this aspect?


**Limitations:**

This is a theoretical work and does not seem to have any potentially negative societal impact.

---

> ### Author Rebuttal · Authors · 2023-08-08
>
> **Comparison to [34] (Question 1 and Weakness 1).**
> The reviewer is right that the analysis in~[34] gives a stronger asymptotic  queue length bound. However, the setting in [34] that resembles our single-queue setting ([34, Theorem 4]) gives a bound with worse dependence on  $\epsilon$. In fact, we show below (at the end of our response) that their bound on $\sum_{t\leq T} Q(t)-Q^{\star}(t)$ is of order $1/\varepsilon^8$, compared to our $\sum_{t\leq T}\frac{Q(t)-Q^\star(t)}{T}\leq \tilde{O}(K/\epsilon)$. In that regard, our bound is stronger for small $T$, whereas the bound in [34] is stronger for large $T$ (even ignoring the gap dependence of their bound). To illustrate that their algorithm has a worse transient performance we simulated the algorithm from [34, Figure 7] under the setting of Fig. 1 in our paper (note that they propose some heuristics for their simulations, though they do not prove theoretical guarantees for these). The figure in the attached PDF shows that their algorithm has a significantly worse transient behavior despite its optimal asymptotic regret scaling. We view this as additional evidence that one should consider a metric focused on early stages, e.g., CLQ.
>
>
> **Technical novelty (Weakness 2).**
> The reviewer is right that our algorithm for the single-queue setting is just UCB; however, the extension to general queueing networks requires a more careful adaptation. Indeed, for learning in general queueing networks, to the best of our knowledge, we provide the first guarantees of any kind. Moreover, our optimal dependence on $\varepsilon$ involves a novel separation of the horizon in two stages (learning and regenerate), which should be of independent interest for analyses of learning in queueing systems. An additional technical benefit of our analysis is that by applying UCB directly, we avoid the inefficiencies induced by forced exploration (which is a common tool for learning in queueing systems [20,34]). Finally, as noted on page 1771 of [34], although forced exploration allows an "easier way to analyze", UCB is "generally a better method and should be used in practice"; and an "interesting direction" is to analyze how UCB can achieve good performance while interacting with the queue's dynamic. Our work serves as an advancement in this direction.
>
>
>
> **Connection to [39] (Question 2).**
>  The reviewer is right that our UCB-based policies may not work in a non-stationary setting. Most closely connected, we highlight that [39] adapts an exponentially decaying rule to the UCB estimation and gives an any-time queue length bound. However, applying their bound in the stationary setting (which is the focus of our paper) does not give a tight dependence on $1/\varepsilon$. As listed in our conclusion (line 365), extending our approach to the nonstationary setting is an exciting future direction.
>
> **Derivation of the $1/\varepsilon^8$ bound for [34, Theorem 4].**
> In the proof of Theorem 4 on page 1770 in [34], the authors shows in Eq. (16) and thereafter that
> $$
> R^{\pi_3}_{(\lambda,\mu)}(T) \leq \sum\_{p=1}^{p_0-1}(M_2p^2+\beta_2) + \sum\_{p = p_0}^T (M_2p^2+\beta_2)M_0e^{-\chi p}
> $$
>
> where $R^{\pi_3}_{(\lambda,\mu)}(T) = \sum\_{t \leq T} (Q^{\pi_3}(t) - Q^{\star}(t))$ and $\pi_3$ is their policy in [34, Figure 7].  The constants $M_0,\chi$ are from Lemma 6 and $M_2,\beta_2$ are from Lemma 8. We obtain lower bounds of these constants from the corresponding proofs as follows.
>
> For $M_0, \chi$, the last inequality (after Eq. (28)) in the proof of Lemma 6 on page 1776 requires $M_0e^{-\chi p}$ to be at least $e^{-\frac{1}{2}p\delta^2} + 2(N-1)e^{-2\frac{p}{N}\gamma^2}$. Here $N$ is the number of servers (i.e., $K$ in our paper), $\gamma$ is the minimum service rate suboptimality gap, and $\delta$ is at most the expected decrease in queue lengths by choosing the fastest server, which is thus at most the traffic slackness $\varepsilon$ in our paper.
>
> For $M_2p^2 + \beta_2$, Eq. (34) in the proof of Lemma 8 on page 1778 requires $M_2 \geq 0$ and $\beta_2 \geq 2\sum_{n=0}^{\infty} n^2e^{-c_3 n}$ and $c_3$ is a constant from Lemma 10. Checking the proof of Lemma 10 on page 1780, after Eq. (41), one can see that $c_3$ is at least $2\delta^2$ and $\delta$ is at most the traffic slackness $\varepsilon$ in our paper (see the definition of $\delta$ after Eq. (40)). Therefore, $\beta_2 \geq \sum_{n=0}^{\infty} n^2e^{-\varepsilon^2 n} = O(\frac{1}{\varepsilon^6})$ since $\int_0^{\infty} x^2e^{-\alpha x} dx = \frac{2}{\alpha^3}$ for any $\alpha > 0$.
>
> Putting these constants together, the upper bound in [34, Theorem 4] is at least (for $T \geq \frac{1}{\varepsilon^2}$)
> $$
> \sum_{p=1}^{p_0-1} (M_2p^2+\beta_2)+\sum_{p=p_0}^T (M_2p^2+\beta_2)M_0e^{-\chi p}  \geq O\left(\sum_{p=1}^{T} \frac{e^{-\frac{1}{2}p\varepsilon^2}}{\varepsilon^6}\right) \geq O\left(\frac{(1-e^{-\frac{1}{2}T\varepsilon^2})}{\varepsilon^8}\right)\geq O\left(\frac{1}{\varepsilon^8}\right).
> $$

---

> > ### Comment · Reviewer_ZtnW · 2023-08-10
> >
> > I thank the authors for their clarification. It is difficult to review a completely new theoretical result at this stage. Even assuming that the authors' claims are correct, compared to [34], they achieve a $\textbf{polynomial}$ improvement w.r.t. $\epsilon$ $(\epsilon^{-8} \to  \epsilon^{-1})$ at the expense of an $\textbf{exponential}$ degradation w.r.t. $T$ ($\log T \to T$), which is hard to justify.
> >
> > In any case, the authors should do a thorough comparison of their results with [34], in particular pointing out that they prove a $\textbf{linear}$ regret bound (which becomes trivial for a reasonably large horizon) compared to a $\textbf{logarithmic}$ regret bound established in [34].

---

> > > ### Author Response · Authors · 2023-08-11
> > >
> > > We thank the reviewer for the response. We want to clarify that the result we provide in the rebuttal is not a new result about our work but the further comparison to [34] that the reviewer requested. The reviewer expressed concern that the algorithm of [34] may be superior to our approach. We show theoretically that this is not the case for the transient behavior (which is the focus of our paper) and display it numerically in a very simple example (in the figure of the attached PDF that we would encourage the reviewer to have another look at as it showcases the high transient queue length in [34]).
> > >
> > > Finally, given that the reviewer's comment doubts the contribution of our work to the literature of learning in queueing systems, we want to reiterate that:
> > >
> > > * The new metric that we propose (CLQ) captures the transient performance of a scheduling algorithm and
> > > can be interpreted as an approximation to the maximum increase in average wait time (see the response to Reviewer KN9r). The reviewer's focus on $T\to\infty$ and the resulting queueing regret does not tackle this important consideration.
> > > * We study much  more general systems, including queueing networks, and propose algorithms whose CLQ matches the provable lower bound. These are vastly more complex (and realistic) systems than the single-queue setting in [34].
> > > * Our bound has no gap dependence, while [34] has (see the discussion in the derivation).
> > >
> > > Of course, we will discuss the comparison in the final version but the reviewer's focus on the single-queue bound for large $T$ ignores that a) our bound is meaningful for the transient setting, b) our approach applies much more generally than the single-queue setting, and c) we make no "gap" assumptions.

---

### Official Review · Reviewer_KN9r · 2023-07-06

**Soundness:** 3 good
**Presentation:** 4 excellent
**Contribution:** 3 good
**Rating:** 7
**Confidence:** 4

**Summary:**

In this paper, the authors propose a new metric to quantify the cost of learning in queueing networks. This notion is required to capture the differences between holding costs in queues and, say costs accumulated in a bandit setting; the latter having a monotonicity property (in expectation). The authors then derive a worst-case lower bound for this metric, and propose UCB based algorithms that come `close' to the lower bound in the order sense.

**Strengths:**

1. The CLQ metric proposed here is meaningful, and quite appropriate for queueing systems. Prior work uses a standard regret metric, which may not be very appropriate given that queue tend to regenerate.

2. The UCB-based analysis appears quite novel; the authors have to bound the CLQ differently in the initial learning phase and subsequently; this issue does not arise in standard UCB analyses for bandits.

3. The analysis extends to queueing networks.

**Weaknesses:**

No significant weaknesses here. I would have liked to see some more exposition in certain places, including a description of the algorithms, and a comparison between the lower and upper bounds derived here. But I can see that the authors have done the best they could to tell the story within the space constraints.

I do have some (minor) suggestions though.

1. I think it is worth highlighting around Theorem 1 that the bound derived is not instance-dependent, but worst-case in nature.

2. The following sentence on Line 270 on Page 7 "Therefore, in expectation, the queue under .... never empties." was unclear to me.


**Questions:**

This is not a serious issue for me, but I did not get what the authors meant in saying that the guarantee on time averaged queue lengths translates to one on wait times via Little's Law. This statement seems somewhat vague. Can CLQ be formally be related to a sub-optimality on average wait times?

**Limitations:**

Not applicable.

---

> ### Author Rebuttal · Authors · 2023-08-08
>
> **Non-instance dependent guarantees (Weakness 1).**
> We appreciate and will adopt the reviewer's suggestion to discuss this further around Theorem 1, and not just in the conclusion (as we currently do).
>
> **Confusing sentence on page 7 (Weakness 2).**
>  We agree that this sentence was confusing; we intended to say that, for any policy, the expected combined service is less than the expected number of arrivals up to this point (which is still confusing); we will try to give a clearer description of this intuition in the camera-ready.
>
> **Connection to suboptimality of wait time (Question 1).**
>  The relationship can be approximately justified through the following derivation. Let $W^{\pi}(T)$ be the average wait time of all jobs up to period $T$ under a policy $\pi$ and $Y(T)$ be the number of arrivals (to all queues) in the first $T$ periods. Then, we obtain from a sample-path version of Little’s Law $$\frac{\sum_{t \leq T} Q^{\pi}(t)}{T} = W^{\pi}(T) * \frac{Y(T)}{T}\text{ and therefore } \frac{\sum_{t \leq T} Q^{\pi}(t)-Q^{\star}(t)}{T} = \left(W^{\pi}(T)-W^{\star}(T)\right) * \frac{Y(T)}{T}.$$ Since $Y(T)$ concentrates near $\lambda T$, the maximum increase in average wait time is approximately $CLQ/\lambda$.

---

### Official Review · Reviewer_aVzT · 2023-07-06

**Soundness:** 4 excellent
**Presentation:** 4 excellent
**Contribution:** 4 excellent
**Rating:** 8
**Confidence:** 4

**Summary:**

This paper studies a problem that involves both learning with queueing.
In the simple setting, there is a single queue served by multiple
servers. However, the service rate at each server is unknown and needs
to be learned. Intuitively, the combination of the learning policy and
the scheduling policy will impact the queue length dynamics of the
system.  Prior work mostly focuses on the queue length performance in
the late stage of the system. However, due to the nature of queueing
systems, this late-stage performance is relatively invariant to the
learning policy, and thus the late-stage metric does not adequately
capture the impact of learning. In contrast, the first contribution of
this paper is to propose a new metric, called CLQ (Cost-of-Learning).
CLQ takes the difference between the time-averaged queue length of a
given policy and that of the optimal, and then takes the maximum over time.
This maximization over the entire time-horizon allows CLQ to capture the
early-stage dynamics of the system, where the impact of learning is more
obvious.  Then, the authors provide both lower bounds and achievable
bounds for this new metrics, which differ only by a logarithmic factor
in the number of servers.  Finally, the results are also extended to
more general multi-queue multi-server systems.

**Strengths:**

1. The new notion of CLQ captures the impact of learning more accurately
than existing studies, by including the early-stage dynamics of the
queue. This is a significant contribution.

2. The lower-bound and achievable-bound are nice and differ only by a
logarithmic factor.

3. The results are extended to general multi-queue multi-server systems.

4. The proof idea based on satisfying regret is also very interesting.

**Weaknesses:**

I don't find major weaknesses.

**Questions:**

1. The work in [39] also provides any-time queueing bounds. I wonder if
they can be translated into an achieveable result for CLQ. Can the
authors comment on how the any-time bound from [39] may compare with the
achieveable bound in this paper?

2. The lower bound involves a very large $K > 2^14 \approx 16000$. Can
the authors comment on what will happen when $K$ is not this large?

Post rebuttal phase:

The reviewer wishes to thank the authors for their response, which clarifies the relation to [39].

**Limitations:**

I do not find discussions on limitations or potential negative societal
impact.

---

> ### Author Rebuttal · Authors · 2023-08-08
>
> **Comparison to [39] (Question 1).**
> The reviewer is right that the guarantee in [39, Theorem 1] can be translated into a CLQ bound. In particular, it implies a cost of learning of $O(\frac{N^4M^4}{\varepsilon^3})$ for a stationary multi-server system with $N$ agents and $M$ workers. This is suboptimal compared with our guarantee (Theorem 4 in the supplementary material), which shows that the cost of learning under \textsc{MW-UCB} is bounded by $O\left(\frac{N^{3.5}M^3}{\varepsilon}\right)$ (see line 123).
>
> **$K$ required in lower bound (Question 2).**
>  The goal of Theorem 1 is to characterize the statistical complexity of learning in queueing and we did not optimize the constant. The exact lower bound we derive (for any $K$) is $\frac{K}{2^{13}\varepsilon}-\frac{1}{2\varepsilon}-1$ (line 522 in the supplementary material); a more careful analysis may be able to reduce the constant in this bound and give a non-vacuous lower bound for small $K$.

---

> > ### Comment · Reviewer_aVzT · 2023-08-11
> >
> > I thank the authors for their response. I will keep my review score.

---

### Official Review · Reviewer_vTq6 · 2023-07-16

**Soundness:** 3 good
**Presentation:** 3 good
**Contribution:** 3 good
**Rating:** 6
**Confidence:** 4

**Summary:**

The authors consider online queuing systems in a discrete time setting. They study settings with single class queue and multi-class queues, and they propose to consider a metric CLQ that serves as a conservative measure on how the queue length(s) could grow across every time point in a horizon. The authors propose a natural UCB algorithm, and demonstrate that it has a near optimal CLQ in a single queue setting. The authors also derive similar bounds in a queueing network setting.

**Strengths:**

- The analysis is quite novel. In addition, the definition of the CLQ and its analysis is new to the best of my knowledge. In particular, the consideration of the satisficing regret is quite interesting.
- The authors achieve a nearly tight result in the case of one queue. Overall, the technical results are solid.
- The notion of CLQ seems adequate as an alternate metric for online queueing systems, but I still have some question (see Question 2 in ``Question'')

**Weaknesses:**

1. The discussions on queueing networks could be benefited from more details. While the authors stated that they will provide additional examples on settings modeled by their queueing networks model in Appendix A, Appendix A does not seem to contain much details. To overcome this weakness, the authors should provide a detail account on how

  - the model on the bipartite queueing system in [11] (Line 187),
  - the model on the multi-server system in [39] (Line 190),

are modeled by the queueing network formulation used by the authors. In particular, it will be crucial for the authors to highlight what are the individual element in the instance tuple (Line 170) in these models. While there could be quite a fair bit of details, I believe that the authors could include them in the Appendix A so that useful details are provided without violating the page limit.

2. There is no simulation to showcase the results.

3. The notion of CLQ seems not to tell us the long term behavior of a policy, since it is taking a worst case over all horizon lengths $1, 2, ....$ (See Question 2)

**Questions:**

- Can I confirm if $\{B_n\}_{n\in{\cal N}}$ is a partition on ${\cal K}$, since the authors say that each server in $B_n$ belongs to one queue?

- While I find the CLQ notion interesting, I still find that it might not have told us the whole story on who the queue length fluctuate under a policy. Let's stick with the single queue model for our discussion sake. For example, let say we have a policy $\pi$ that achieve a CLQ of $\text{constant} \times K/\epsilon$, which is optimal, within a constant factor. Knowing that $\mathbf{E}[Q(t, \pi^*)] = O(1/\epsilon)$ for a large $t$ (I think you can show this by a similar logic to how we derive the average queue length of a stable M/M/1), achieving an optimal CLQ stated above actually does not prevent us from having $\mathbf{E}[Q(t, \pi)] = O(K/\epsilon)$ for all sufficiently large $t$. In this way, achieving an optimal CLQ does not necessarily mean that we converge to the optimal queue length when $t$ grows.

A priori, this might not be a critical issue since the authors aspire to investigate a short time horizon regime and not only a long one, and IMHO I regard CLQ as a conservative measure that tells us how long the queue length could be over all possible horizon lengths.  But I am wondering if the authors could also consider a worst case over the horizon lengths in $\{\tau, \tau+1, \ldots \}$ instead of $\{1, 2, \ldots \}$, for an arbitrary $\tau$? More precisely, do the authors' analysis inform us of a bound on (3) with the range of maximum $\{1, 2, \ldots \}$ replaced with $\{\tau, \tau+1, \ldots \}$ for and arbitrary $\tau$?

- What is the difference between the two definitions (5), (6)?


**Limitations:**

There is no potential societal impact to my best knowledge. This is a theoretical paper.

---

> ### Author Rebuttal · Authors · 2023-08-08
>
> **Long-term optimality (Question 2/Weakness 3).**
> We thank the reviewer for pointing out that we may get a better bound for a larger $\tau$. Indeed, our analysis directly extends to show that UCB converges to the optimal $O(1/\varepsilon)$ scaling (for the single-queue setting). By combining Lemma 4.2 and 4.3 we can obtain a bound of $\tilde{O}(1/\varepsilon + K^2/(T \varepsilon^4))$ for the time-averaged queue length in the initial $T$ periods. As a result, if $\tau \geq K^2/\varepsilon^4$, the maximum of time-averaged queue lengths for $T \geq \tau$ will be bounded by $O(1/\varepsilon)$ with no dependence on $K$.
>
> **Clarification comments (Question 1, Question 3, and Weakness 2).**
> The reviewer is right that $\\{\mathcal{B}_n\\}$ is a partition of $\mathcal{K}$. The difference between (5) and (6) is that (5) is tailored to the multi-queue multi-server system, i.e., the DM knows that there are no queue transitions; this is a special case of the queueing networks whose CLQ we define in (6). The reviewer is right that we do not have a subsection for simulation results, but we highlight that Figure 1 includes some numerical evidence for the performance of our algorithm (see also the PDF uploaded with an updated figure).
>
> **More details on the queueing models (Weakness 1).** In a bipartite queueing system [11], there are $N$ agents and $M$ workers. In period $t$, a new job arrives to each agent $n$’s queue with probability $\lambda_n$. The DM selects a matching $\sigma_{n,m}$ between agents and workers such that $\sum_{m’\in[M]} \sigma_{n,m’}\leq 1,\sum_{n’\in [N]} \sigma_{n’,m} \leq 1, \forall n \in [N], m\in [M]$. If $\sigma_{n,m}=1$, then the first job in agent $n$’s queue (if any) is cleared with probability $\mu_{n,m}$. Otherwise, the job stays in the queue. All arrivals and services are independent across queues, servers and periods. To translate this model into our queueing network formulation (lines 168-204), let $\mathcal{N} = [N], \mathcal{K} = [N] \times [M] = \\{(n,m), n\in [N],m\in [M]\\}$. $\Lambda$ corresponds to $N$ independent Bernoulli random variables. For each $k=(n,m) \in \mathcal{K}$, we have $\mu_k = \mu_{n,m}$. The set $\mathcal{A} = \\{0,1\\}^{\mathcal{N}}$.
> $\Sigma$ is the set of subsets $\boldsymbol{\sigma}$ of $\mathcal{K}$, such that $\sum_{k=(n,m’)} \sigma_k \leq 1, \sum_{k=(n’,m)} \sigma_k \leq 1, \forall n\in[N],m\in[M]$. $\mathcal{B}_n = \\{k=(n,m), m \in [M]\\}$. $\mathcal{D}_n = \emptyset$ for all $n$, and $P$ only includes transitions to the virtual queue.
>
> The multi-server system [39] is similarly defined but instead of selecting a matching between agents and workers, the DM can match an agent with multiple workers, as long as there are sufficiently many jobs assigned to different workers. Therefore, the only change compared with the bipartite queueing system would be that $\Sigma$ consists of all $\boldsymbol{\sigma}$ such that $\sum_{k=(n’,m)} \sigma_k \leq 1, \forall m \in [M]$.

---

### Author Rebuttal · Authors · 2023-08-08

We conducted a new simulation with the algorithm from [34, Figure 7] under the same setting of Figure 1 in our paper. The figure in the attached PDF shows that their algorithm has a significantly worse transient behavior despite its optimal asymptotic regret scaling.

---

### Decision · Program_Chairs · 2023-09-21

**Decision:**

Accept (poster)

**Comment:**

The paper investigates a queueing system where the service rates of the servers are initially unknown. The servers at a given queue are not working in parallel (which greatly simplifies the analysis), and corresponds more to a channel selection problem in radio communication systems. The authors introduce the CLQ metric defined as the maximum over time of the empirical expected simple regret. The latter is the difference between the queue length under the considered algorithm and the oracle algorithm (knowing the servers' rates). The authors establish a lower bound on the CLQ valid only for specific systems, and prove that UCB achieves this lower bound when it comes to the scaling of the slackness parameter (the distance to having an unstable system -- heavy traffic regime). These contributions are acknowledged by the reviewers.

However there are serious concerns about the way the related work is treated in the paper, and the connection to [20] and [34]. It is true that [34] and the current submission do not look at the same metric, as [34] is mainly concerned with the asymptotic regime when $T$ grows large. And the CLQ metric is of interest because it captures the transient behavior of the learning process. Nevertheless, the authors should explain the differences in the paper in more detail.

An exhaustive comparison with [20] seems even more important. In [20], the authors study both the asymptotic regime as $T$ grows large, and the transient regime in heavy traffic. The current submission only addresses the latter regime. Most of the qualitative observations made in the current submission are made in [20]: https://arxiv.org/pdf/1604.06377.pdf. See e.g. Fig. 1, Sections 5 and 6.

-	In particular, in Section 5 of [20], the authors study the early stage regime, and they already prove a lower bound. The authors should compare it to their lower bound presented in Theorem 1. To be more precise, in Proposition 7, [20] establishes that on an interval of the type $[a,b/\epsilon]$, the regret per round is greater than $K\log(t)/\log\log(t)$. This seems like a very precise result. Fortunately, as it seems, the lower bound presented in Theorem 1 cannot be deduced from this result. The lower bound on the CLQ you would obtain from Proposition 7 in [20] would be much smaller than that in the current submission. Note however that the lower bound presented in Theorem 1 is valid only for seepcific cases, for $K\ge 2^{14}$.

-	Section 6 [20] presents nice numerical results illustrating the early and late stages. [20] already noticed that UCB was better in early stages.

-	In [20], the regret upper bound of Q-UCB for the early stage scales as $1/\epsilon^2$, not $1/\epsilon$ as it should and as the UCB algorithm studied in the present submission. On the other hand, UCB has worse performance in the late stage.

The paper has in view the above discussion interesting results compared to [20], but the authors should provide a proper and fair comparison with [20].

An additional remark is that the CLQ metric is defined as a maximum over time, and hence it does not tell us when the CLQ value is achieved. If you wish to focus your attention on the early stages, then looking at the evolution of the simple regret is more precise. This is what is done in [20]. There, the authors give the shape of the evolution of the simple regret (see their Figure 1).